# Indirect reciprocity with stochastic and dual reputation updates

Yohsuke Murase[1,2]*, Christian Hilbe[2]

**1** RIKEN Center for Computational Science, Kobe, Japan, **2** Max Planck Research Group 'Dynamics of Social Behavior', Max Planck Institute for Evolutionary Biology, Plön, Germany

* yohsuke.murase@gmail.com

**Data Availability Statement:** The source code is available at https://github.com/yohm/sim_indirect_recip_stochastic.

**Funding:** Y.M. acknowledges support from Japan Society for the Promotion of Science (JSPS) (JSPS KAKENHI; Grant no. 21K03362, Grant no.

## Abstract

Cooperation is a crucial aspect of social life, yet understanding the nature of cooperation and how it can be promoted is an ongoing challenge. One mechanism for cooperation is indirect reciprocity. According to this mechanism, individuals cooperate to maintain a good reputation. This idea is embodied in a set of social norms called the "leading eight". When all information is publicly available, these norms have two major properties. Populations that employ these norms are fully cooperative, and they are stable against invasion by alternative norms. In this paper, we extend the framework of the leading eight in two directions. First, we include norms with 'dual' reputation updates. These norms do not only assign new reputations to an acting donor; they also allow to update the reputation of the passive recipient. Second, we allow social norms to be stochastic. Such norms allow individuals to evaluate others with certain probabilities. Using this framework, we characterize all evolutionarily stable norms that lead to full cooperation in the public information regime. When only the donor's reputation is updated, and all updates are deterministic, we recover the conventional model. In that case, we find two classes of stable norms: the leading eight and the 'secondary sixteen'. Stochasticity can further help to stabilize cooperation when the benefit of cooperation is comparably small. Moreover, updating the recipients' reputations can help populations to recover more quickly from errors. Overall, our study highlights a remarkable trade-off between the evolutionary stability of a norm and its robustness with respect to errors. Norms that correct errors quickly require higher benefits of cooperation to be stable.

## Author summary

Indirect reciprocity is a mechanism for cooperation based on social norms. The respective field explores which norms are stable, and how changes in a social norm affect a population's cooperation rate. Here, we generalize a classical framework of indirect reciprocity in two ways. First, we allow for norms in which also the reputations of passive receivers are updated. Second, we allow social norms to have a stochastic component. We use this general framework to characterize all cooperative norms that are evolutionarily stable.

21KK0247, Grant no. 22H00815). C.H. acknowledges generous funding from the European Research Council (ERC) under the European Union's Horizon 2020 research and innovation program (Starting Grant 850529: E-DIRECT), and from the Max Planck Society. The funders had no role in study design, data collection and analysis, decision to publish, or preparation of the manuscript.

**Competing interests:** The authors have declared that no competing interests exist.

Our results provide a new perspective for understanding previous findings, and they can serve as a foundation for future studies in this area.

## Introduction

Humans have a remarkable ability to cooperate with others [1–3]. This ability is particularly striking in social dilemmas, in which individuals cooperate despite any immediate incentives to defect. Such cooperative interactions can be rationalized when they happen in public. In that case, cooperation may help individuals to gain a good reputation, which in turn may be valuable in future interactions. This logic of public cooperation is the basic premise of models of indirect reciprocity [4–7].

In models of indirect reciprocity, the interplay between an individual's actions and the resulting reputations is governed by a community's social norm. Social norms can be conceptualized as a combination of an assessment rule and an action rule [8]. The assessment rule determines how reputations are assigned to community members, depending on who did what to whom. The norm's action rule determines how people should act, which may depend on their own reputation and the reputation of their interaction partner. The aim of studies in indirect reciprocity is to find those social norms that lead to stable cooperation.

One set of such norms are the so-called 'leading eight', which were discovered by Ohtsuki and Iwasa [9, 10]. The leading eight achieve full cooperation in the limit of low error rates. In addition, they form a strict Nash equilibrium when error rates are small but positive. We refer to norms with those two properties as cooperative ESS (CESS). The leading eight's assessment rules and action rules are illustrated in Table 1. They can be summarized with the following four principles: (*i*) Maintenance of cooperation: When good donors encounter a good recipient, they should cooperate. This in turn should preserve the donors' good reputation. (*ii*) Identification of defectors: Donors who defect against good recipients should be deemed as bad. (*iii*) Justified punishment: When good donors encounter a bad recipient they should defect, without harm to their good reputation. (*iv*) Apology and forgiveness: Bad donors should cooperate when they meet a good recipient; in return they should recover a good reputation. Because the leading eight are both simple and effective, they have become the main reference

**Table 1. The prescriptions of the leading eight.** The top row (*X, Y*) indicates the reputations of the donor and the recipient, respectively. For instance, (*G, B*) means the case where a good (*G*) donor meets a bad (*B*) recipient. The rules $P$, $R_1(C)$, $R_1(D)$ indicate the prescribed action, the assessment when cooperation (*C*) is observed, and the assessment when defection (*D*) is observed, respectively. The entry 1 means *C* or *G*, whereas 0 means *D* or *B*. The entries that are different from each other are highlighted in bold text. In the bottom row, the common prescriptions are shown, where * means that the entry is arbitrary and † means that the entry is determined according to $R_1(B, B, C)$ and $R_1(B, B, D)$. These norms are CESS for $b/c > 1$ in the rare error limit.

| | (G, G) | | | (G, B) | | | (B, G) | | | (B, B) | | |
|---|---|---|---|---|---|---|---|---|---|---|---|---|
| | $P$ | $R_1(C)$ | $R_1(D)$ | $P$ | $R_1(C)$ | $R_1(D)$ | $P$ | $R_1(C)$ | $R_1(D)$ | $P$ | $R_1(C)$ | $R_1(D)$ |
| *L*1 | 1 | 1 | 0 | 0 | **1** | 1 | 1 | 1 | 0 | **1** | **1** | **0** |
| *L*2 (Consistent Standing) | 1 | 1 | 0 | 0 | **0** | 1 | 1 | 1 | 0 | **1** | **1** | **0** |
| *L*3 (Simple Standing) | 1 | 1 | 0 | 0 | **1** | 1 | 1 | 1 | 0 | **0** | **1** | **1** |
| *L*4 | 1 | 1 | 0 | 0 | **1** | 1 | 1 | 1 | 0 | **0** | **0** | **1** |
| *L*5 | 1 | 1 | 0 | 0 | **0** | 1 | 1 | 1 | 0 | **0** | **1** | **1** |
| *L*6 (Stern Judging) | 1 | 1 | 0 | 0 | **0** | 1 | 1 | 1 | 0 | **0** | **0** | **1** |
| *L*7 (Staying) | 1 | 1 | 0 | 0 | **1** | 1 | 1 | 1 | 0 | **0** | **0** | **0** |
| *L*8 (Judging) | 1 | 1 | 0 | 0 | **0** | 1 | 1 | 1 | 0 | **0** | **0** | **0** |
| common | 1 | 1 | 0 | 0 | * | 1 | 1 | 1 | 0 | † | * | * |

for many subsequent theoretical studies, including studies on costly punishment [11], incomplete and private observations [12, 13], ingroup favoritism [14, 15], and private reputations [16–27] to name a few [28–41].

In this paper, we extend this previous work in two ways. Our first extension addresses the reputations of recipients. In previous studies, social norms only determined how the donor's reputation is updated. In contrast, the recipient's reputation was kept constant. However, previous empirical work as well as everyday experience suggest that some social interactions also affect the reputations of passive receivers. For instance, there is some experimental evidence that individuals are sympathetic to the victims of a defector [42]. To explain such regularities, one may consider social norms in which recipients of a defecting donor deserve a good reputation. Conversely, there may also be social norms that have the opposite effect. When observing a donor who refuses to help a recipient, bystanders may infer that the recipient must have had a bad reputation to begin with. Our "dual reputation update" framework allows us to study such examples more thoroughly. Remarkably, we find that the respective norms can be more resilient with respect to errors.

Our second extension explores the role of stochasticity in social norms. Most previous studies presume social norms to be deterministic. Deterministic norms have the formal advantage that they can be enumerated, and hence they can be studied exhaustively. However, in reality, actions and assessments may not need to be deterministic for various reasons. For instance, assessments may well depend on additional factors that are exogenous to the model, such as an individual's mood, or the weather. Stochasticity is one way to take into account such factors without increasing a norm's complexity. Recently, stochastic rules received some attention in models of private reputations [43, 44]. In particular, Schmid et al. suggested a stochastic norm of Generous Scoring [43]. This norm is both cooperative and it forms a (non-strict) Nash equilibrium. We contribute to this line of research by exploring stochastic norms in a framework of public reputations. This approach allows us to derive the necessary and sufficient conditions for such a norm to be a CESS. Our model includes the deterministic norms as special cases. As a consequence, we recover the deterministic CESS norms, including the leading eight in the respective limit. In this way, our results offer a new perspective on existing results. In addition, we identify cases in which stochasticity can further promote the stability of cooperation.

This paper is organized as follows. In the next section, we introduce the model. The subsequent Results section consists of three subsections. The first subsection explores how social norms affect the reputation dynamics within a population. The second subsection builds on these results to derive the necessary and sufficient conditions for a norm to be a CESS. In the third subsection, we apply these general results to investigate several special cases, including deterministic norms with and without updating the recipient's reputation. The last section provides a summary and a discussion.

## Model

In this study, we follow the basic framework of Ohtsuki and Iwasa [9]. We consider an infinitely large population of players who interact in pairwise donation games. In each round, two players are randomly chosen as a donor and a recipient. The donor decides whether to cooperate ($C$) or to defect ($D$). Cooperation incurs a cost $c > 0$ on the donor and results in a benefit $b > c$ for the recipient. Defection leads to a payoff of zero for both players. If the donation game is only played once, the donor is better off by defecting, creating a social dilemma. However, we consider the case that members of the population play many donation games against different opponents. In that case, individuals can build up a reputation over time, which may affect

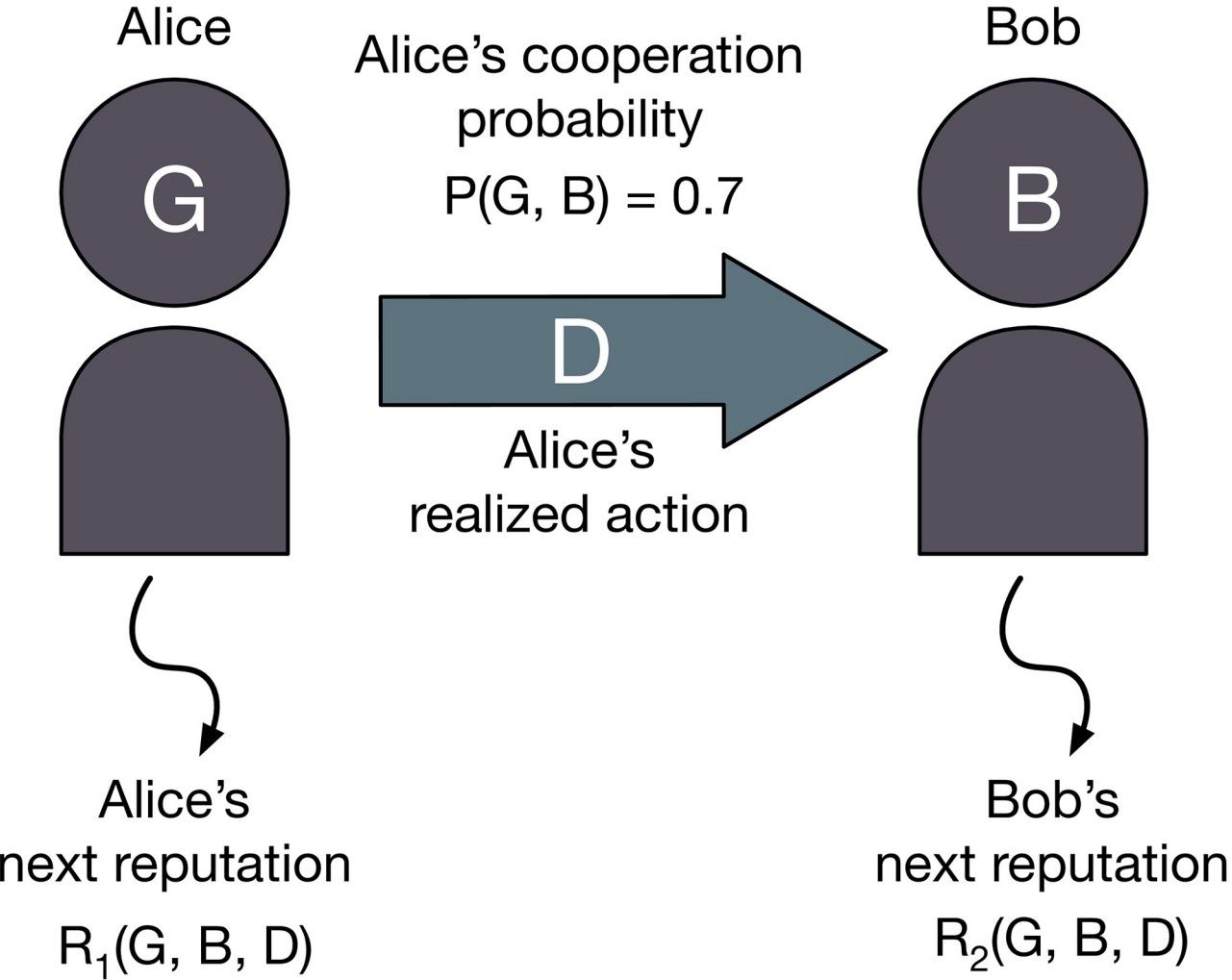

**Fig 1. A schematic diagram of the model.** At each time step, a donor and a recipient are randomly selected from the population. Let us suppose the donor (Alice) and the recipient (Bob) have reputations $G$ and $B$, respectively. Alice decides her action according to her action rule $P$, which returns the probability of cooperation. In this example, the cooperation probability is $P(G, B) = 0.7$, and the realized action is $D$. The reputation of Alice and Bob are updated according to the assessment rules $R_1$ and $R_2$. Alice and Bob get $G$ reputation with probability $R_1(G, B, D)$ and $R_2(G, B, D)$, respectively. This process repeats changing the donor-recipient pairs.

how individuals cooperate. Here, we assume that reputations are binary. Players are either good ($G$) or bad ($B$).

How players form reputations, and how they act based on these reputations, depends on their social norm. In our study, a social norm consists of an action rule and two assessment rules, as shown in Fig 1. The action rule $P(X, Y)$ determines whether a player should cooperate or defect when acting as a donor. This choice might depend on the player's own reputation $X$ as well as on the reputation $Y$ of the recipient, where $X, Y \in \{G, B\}$. The output $P(X, Y) \in [0, 1]$ is the probability with which the donor cooperates. In addition, we consider two assessment rules. The first rule, $R_1(X, Y, A)$ is the probability that the donor is assigned a good reputation after the interaction. This probability depends on the previous reputation $X$ of the donor, the previous reputation $Y$ of the recipient, and on the donor's action $A \in \{C, D\}$ in the donation game. The second assessment rule $R_2(X, Y, A)$ is the probability that the recipient is assigned a

good reputation after the interaction. When the output of all three rules $P(X, Y)$, $R_1(X, Y, A)$, $R_2(X, Y, A)$ is constrained to be either zero or one, the norm is deterministic; otherwise we call it stochastic.

Both the players' actions and their assessments can be subject to errors. First, we consider implementation errors. Such errors affect donors who wish to cooperate; with probability $\mu_e$ such donors defect by mistake. As a result, instead of their intended strategy $P(X, Y)$, players implement the effective strategy

$$\tilde{P}(X, Y) = (1 - \mu_e)P(X, Y). \tag{1}$$

Second, we consider assessment errors. These errors happen when new reputations are assigned to the donor and the recipient. With probabilities $\mu_{a1}$ and $\mu_{a2}$, the respective assignments are the opposite of the assignment prescribed by the social norm. As a result, the effective assessment rules become

$$\tilde{R}_1(X, Y, A) = (1 - \mu_{a1})R_1(X, Y, A) + \mu_{a1}[1 - R_1(X, Y, A)] \tag{2}$$

and

$$\tilde{R}_2(X, Y, A) = (1 - \mu_{a2})R_2(X, Y, A) + \mu_{a2}[1 - R_2(X, Y, A)]. \tag{3}$$

In the presence of these errors, we obtain the constraints $0 \leq \tilde{P}(X, Y) \leq (1 - \mu_e)$, $\mu_{a1} \leq \tilde{R}_1(X, Y, A) \leq 1 - \mu_{a1}$, and $\mu_{a2} \leq \tilde{R}_2(X, Y, A) \leq 1 - \mu_{a2}$. An important special case arises when the recipients' reputation is kept constant, $R_2(X, Y, A) = \delta_{Y,G}$ and $\mu_{a2} = 0$, where $\delta_{Y,G}$ is the Kronecker delta function. If norms are additionally assumed to be deterministic, we recover the model by Ohtsuki and Iwasa [9].

In agreement with Ohtsuki and Iwasa's study, we consider a public information model. That is, all players learn the same information and share the same assessment of any given population member at any point in time. These shared assessments can change in time, depending on the population members' interactions. Herein, we assume that players interact in sufficiently many donation games such that a player's reputation assignments reach a stationary state.

In the remainder of this article, we are interested in which social norms are CESS. The respective norms have two properties. First, they are *self-cooperative*: if the norm is adopted by everyone, the population's cooperation rate approaches one in the limit of rare errors. Second, for positive but small error rates, we require the respective norm to form a strict Nash equilibrium. That is, if an infinitesimal minority of the population adopts a different norm, the minority yields a lower payoff than the residents.

## Results

### Description of the reputation dynamics

To characterize the CESS, we first need to describe how reputations change in time, depending on which social norm the players adopt. To this end, we first assume that everyone in the population adopts the same social norm. In a second step, we discuss the case in which a small minority of players switches to a different norm.

**Reputation dynamics in homogeneous populations.** Consider a homogeneous population that uses the action rule $P(X, Y)$, and the assessment rules $R_1(X, Y, A)$ and $R_2(X, Y, A)$. At any given time $t$, let $h(t)$ denote the fraction of players with a good reputation. Similarly, $1 - h(t)$ is the fraction of players with bad reputation. Then $h(t)$ obeys the following differential

equation,

$$
\begin{aligned}
\dot{h}(t) \;=\;& h(t)^2[\overline{R}_1(G,G) + \overline{R}_2(G,G)] \\
+\;& h(t)\,(1-h(t))[\,\overline{R}_1(G,B) + \overline{R}_2(G,B) + \overline{R}_1(B,G) + \overline{R}_2(B,G)] \\
+\;& (1-h(t))^2\,[\,\overline{R}_1(B,B) + \overline{R}_2(B,B)] \\
-\;& 2h(t).
\end{aligned}
\tag{4}
$$

In this expression, $\overline{R}_1(X,Y)$ and $\overline{R}_2(X,Y)$ refer to the expected probabilities to assign a good reputation to the donor and the recipient if their initial reputations are $X$ and $Y$, respectively. These expected probabilities are defined as

$$
\begin{aligned}
\overline{R}_1(X,Y) &\equiv \tilde{P}(X,Y)\tilde{R}_1(X,Y,C) + (1 - \tilde{P}(X,Y))\tilde{R}_1(X,Y,D) \\
\overline{R}_2(X,Y) &\equiv \tilde{P}(X,Y)\tilde{R}_2(X,Y,C) + (1 - \tilde{P}(X,Y))\tilde{R}_2(X,Y,D).
\end{aligned}
\tag{5}
$$

The last term of Eq (4) has a coefficient of $-2$ because both the donor's and the recipient's reputation is updated.

As $t \to \infty$, the proportion of good population members $h(t)$ converges to a fixed point $h^* \in [0, 1]$. This fixed point is unique and stable, because the above equation is quadratic with respect to $h$ and because $\dot{h}|_{h=1} < 0$ and $\dot{h}|_{h=0} > 0$ when $\mu_a > 0$. The stationary value is obtained by solving the quadratic equation

$$
Ah^{*2} + Bh^* + C = 0,
\tag{6}
$$

where $A$, $B$, and $C$ are defined as

$$
\left\{
\begin{aligned}
A \;\equiv\;& \overline{R}_1(G,G) + \overline{R}_2(G,G) - \overline{R}_1(G,B) - \overline{R}_2(G,B) \\
& - \overline{R}_1(B,G) - \overline{R}_2(B,G) + \overline{R}_1(B,B) + \overline{R}_2(B,B) \\
B \;\equiv\;& \overline{R}_1(G,B) + \overline{R}_2(G,B) + \overline{R}_1(B,G) + \overline{R}_2(B,G) \\
& - 2\overline{R}_1(B,B) - 2\overline{R}_2(B,B) - 2 \\
C \;\equiv\;& \overline{R}_1(B,B) + \overline{R}_2(B,B).
\end{aligned}
\right.
\tag{7}
$$

The unique solution $h^* \in [0, 1]$ to the quadratic Eq (6) is

$$
h^* =
\begin{cases}
\dfrac{-B - \sqrt{B^2 - 4AC}}{2A} & \text{when } A \neq 0 \\[2ex]
-\dfrac{C}{B} & \text{when } A = 0.
\end{cases}
\tag{8}
$$

At the stationary state, the probability that a donor cooperates is then

$$
p_{\text{res}\to\text{res}} = h^{*2}\tilde{P}(G,G) + h^*(1-h^*)(\tilde{P}(G,B) + \tilde{P}(B,G)) + (1-h^*)^2\tilde{P}(B,B).
\tag{9}
$$

In particular, for the social norm to be self-cooperative, this expression needs to approach one for small error rates.

**Reputation dynamics in populations with rare mutants.** To explore whether a social norm is stable, we need to explore whether players have an incentive to deviate. This in turn depends on the reputational consequences of a deviation. To explore these consequences in more detail, we again consider a resident population with action rule $P(X, Y)$ and assessment

rules $R_1(X, Y, A)$ and $R_2(X, Y, A)$. Suppose now there is also an infinitesimal number of mutant players. These mutants deviate by following a different action rule $P'(X, Y)$. We do not study deviations in the assessment rule because rare mutants have no influence on how the population assigns reputations, see Ref. [9]. As before, the effective action rule of a mutant is

$$\tilde{P}'(X, Y) = (1 - \mu_e)P(X, Y). \tag{10}$$

Similar to before, let $H(t)$ denote the fraction of mutants with good reputation. When the resident population is at the steady state, $H(t)$ evolves as follows,

$$
\begin{aligned}
\dot{H}(t) \quad &= h^* H(t)\overline{R}_1(G, G|P') \\
&+ h^* H(t)\overline{R}_2(G, G|P) \\
&+ h^*(1 - H(t))\overline{R}_1(B, G|P') \\
&+ h^*(1 - H(t))\overline{R}_2(G, B|P) \\
&+ (1 - h^*)H(t)\overline{R}_1(G, B|P') \\
&+ (1 - h^*)H(t)\overline{R}_2(B, G|P) \\
&+ (1 - h^*)(1 - H(t))\overline{R}_1(B, B|P') \\
&+ (1 - h^*)(1 - H(t))\overline{R}_2(B, B|P) \\
&- 2H(t).
\end{aligned}
\tag{11}
$$

The last term has the coefficient $-2$ because the mutant is subject to change in reputation either as a donor or as a recipient. In the above equation, we used the following notation for the expected reputation of the donor and the recipient,

$$
\begin{cases}
\overline{R}_1(X, Y|P') &\equiv \tilde{P}'(X, Y)\tilde{R}_1(X, Y, C) + (1 - \tilde{P}'(X, Y))\tilde{R}_1(X, Y, D) \\
\overline{R}_2(X, Y|P) &\equiv \overline{R}_2(X, Y)
\end{cases}
. \tag{12}
$$

Eq (11) describes the case that a mutant meets a resident either as a donor or a recipient. We do not need to take into account cases in which a mutant meets another mutant because mutants are infinitesimally rare. After a sufficiently long time, $H(t)$ converges to the unique stable fixed point

$$H^* = \frac{h^* H_1 + (1 - h^*)H_2}{2 - h^* H_3 - (1 - h^*)H_4}, \tag{13}$$

where

$$
\begin{cases}
H_1 \equiv \overline{R}_1(B, G|P') + \overline{R}_2(G, B|P) \\
H_2 \equiv \overline{R}_1(B, B|P') + \overline{R}_2(B, B|P) \\
H_3 \equiv \overline{R}_1(G, G|P') + \overline{R}_2(G, G|P) - \overline{R}_1(B, G|P') - \overline{R}_2(G, B|P) \\
H_4 \equiv \overline{R}_1(G, B|P') + \overline{R}_2(B, G|P) - \overline{R}_1(B, B|P') - \overline{R}_2(B, B|P)
\end{cases}
. \tag{14}
$$

Using these stationary values, the cooperation probability of a mutant against a resident becomes

$$
\begin{aligned}
p_{\text{mut}\to\text{res}} \quad &= H^* h^* \tilde{P}'(G, G) + H^*(1 - h^*)\tilde{P}'(G, B) \\
&+ (1 - H^*)h^* \tilde{P}'(B, G) + (1 - H^*)(1 - h^*)\tilde{P}'(B, B)
\end{aligned}
. \tag{15}
$$

Conversely, the cooperation probability of a resident against a mutant is

$$
\begin{aligned}
p_{\mathrm{res}\to\mathrm{mut}} \quad &= H^* h^* \tilde{P}(G, G) + H^*(1 - h^*)\tilde{P}(B, G) \\
&\quad + (1 - H^*)h^* \tilde{P}(G, B) + (1 - H^*)(1 - h^*)\tilde{P}(B, B)
\end{aligned}
\tag{16}
$$

Therefore, the payoffs of the resident and the mutant are

$$
\begin{cases}
\pi_{\mathrm{res}} = (b - c)\, p_{\mathrm{res}\to\mathrm{res}} \\
\pi_{\mathrm{mut}} = b\, p_{\mathrm{res}\to\mathrm{mut}} - c\, p_{\mathrm{mut}\to\mathrm{res}}.
\end{cases}
\tag{17}
$$

The resident is strictly stable against the mutant when $\pi_{\mathrm{res}} > \pi_{\mathrm{mut}}$, that is, when

$$
\begin{cases}
\dfrac{b}{c} > \dfrac{p_{\mathrm{res}\to\mathrm{res}} - p_{\mathrm{mut}\to\mathrm{res}}}{p_{\mathrm{res}\to\mathrm{res}} - p_{\mathrm{res}\to\mathrm{mut}}} & \text{if } p_{\mathrm{res}\to\mathrm{res}} > p_{\mathrm{res}\to\mathrm{mut}} \\[2mm]
\dfrac{b}{c} < \dfrac{p_{\mathrm{res}\to\mathrm{res}} - p_{\mathrm{mut}\to\mathrm{res}}}{p_{\mathrm{res}\to\mathrm{res}} - p_{\mathrm{res}\to\mathrm{mut}}} & \text{if } p_{\mathrm{res}\to\mathrm{res}} < p_{\mathrm{res}\to\mathrm{mut}} \\[2mm]
p_{\mathrm{res}\to\mathrm{res}} < p_{\mathrm{mut}\to\mathrm{res}} & \text{if } p_{\mathrm{res}\to\mathrm{res}} = p_{\mathrm{res}\to\mathrm{mut}}
\end{cases}
\tag{18}
$$

A social norm is an ESS when the resident is strictly stable against all possible mutants. In the following, we describe all norms that are an ESS and self-cooperative.

## A characterization of CESS norms

For our analysis, we assume vanishing error rates, $\mu_e \to 0^+$, $\mu_{a1} \to 0^+$, and $\mu_{a2} \to 0^+$. In this limit, effective rules converge to the original rules, $\tilde{P} \to P$, $\tilde{R}_1 \to R_1$ and $\tilde{R}_2 \to R_2$.

**Self-cooperative norms.** To start with, we first describe which norms are self-cooperative, such that $p_{\mathrm{res}\to\mathrm{res}} = 1$. Before we go into detail, let us first show that for any such CESS norm either $h^* = 1$ or $h^* = 0$ must hold. To see why, assume to the contrary that $0 < h^* < 1$, such that there are both good and bad players in the population. For the norm to be self-cooperative, the action rule needs to prescribe cooperation in all possible cases. Therefore, $P(G, G) = P(G, B) = P(B, G) = P(B, B) = 1$. However, such a norm of unconditional cooperation is not an ESS because it can be invaded by unconditional defectors. As the two labels $G$ and $B$ are interchangeable, we consider without loss of generality the case that $h^* = 1$ in the following. That is, when the respective social norm is adopted by the entire population, we assume everyone is assigned a good reputation eventually.

To have $h^* = 1$, the following conditions are necessary and sufficient:

$$
h^* = 1 \Leftrightarrow
\begin{cases}
\dot{h}\big|_{h=1} = 0 \\[2mm]
\dfrac{d\dot{h}}{dh}\bigg|_{h=1} < 0
\end{cases}
\tag{19}
$$

The upper equation on the right hand side makes sure that there is a fixed point at $h = 1$. The second inequality indicates that this fixed point is stable. By Eq (4) these two requirements are equivalent to the following conditions,

$$
\begin{cases}
\overline{R}_1(G, G) = 1 \\
\overline{R}_2(G, G) = 1 \\
\overline{R}_1(G, B) + \overline{R}_2(G, B) + \overline{R}_1(B, G) + \overline{R}_2(B, G) > 2.
\end{cases}
\tag{20}
$$

In addition, given $h^* = 1$, we can use Eq (9) to conclude that the social norm is self-cooperative, $p_{\mathrm{res}\to\mathrm{res}} = 1$, if and only if

$$P(G, G) = 1. \tag{21}$$

We conclude that the self-cooperative norms in which all population members have a good reputation are exactly those that satisfy conditions (20) and (21).

**Evolutionary stability of self-cooperative norms.** Next, we derive some necessary conditions for a self-cooperative norm to be an ESS. Without loss of generality, we restrict attention to deterministic action rules $P(X, Y)$, because for any given context $(X, Y)$ it is either optimal to always cooperate or to never do so. When the expected long-term payoff for cooperation is higher than that for defection, $P(X, Y) = 1$ is the optimal action, and vice versa. We note that in some contexts $(X, Y)$ and for specific $b/c$ values, cooperation and defection may yield identical long-term payoffs. In that case, any stochastic rule with $P(X, Y) \in (0, 1)$ also yields the same payoff. However, because all of the respective norms are neutral with respect to one another, none of them is evolutionarily stable.

As a first requirement for a social norm to be a CESS, we note that a good donor must defect against a bad recipient,

$$P(G, B) = 0. \tag{22}$$

To see why, we observe that $h^* = 1$ implies that both good and bad players are almost always matched with good players. If good players cooperate with bad players, i.e., $P(G, B) = 1$, both good and bad players would fully receive the benefit of cooperation irrespective of their reputations. But in that case, reputations would be inconsequential, and players would have no incentive to cooperate to begin with. Therefore, such a social norm cannot be a CESS.

To make further progress, in the following we consider the cases $P(B, G) = 1$ and $P(B, G) = 0$ separately. In each case, we check whether the given action rule $P(X, Y)$ is optimal, for each possible context that the donor's and the recipient's reputation are either $(G, G)$, $(B, G)$, $(G, B)$, or $(B, B)$. From these comparisons, we obtain necessary and sufficient conditions for a social norm to be a CESS.

**Norms with $P(B, G) = 1$.** (*i*) To explore whether the given action rule is optimal, let us first consider the context that a bad donor is matched with a good recipient, $(B, G)$. If the donor acts according to the resident strategy, its action rule prescribes to cooperate $P(B, G) = 1$. If the donor acts according to the mutant strategy, then $P'(B, G) = 0$. Since the donor starts with a bad reputation, the expected number of rounds until the donor gets a good reputation is

$$\begin{cases} T &= \dfrac{2}{R_1(B, G, C) + R_2(G, B, D)} \\[2mm] T' &= \dfrac{2}{R_1(B, G, D) + R_2(G, B, D)}, \end{cases} \tag{23}$$

for the resident and the mutant, respectively. We note that $T' > T$ is required, for otherwise the norm cannot be an ESS. Thus, $R_1(B, G, C) > R_1(B, G, D)$ is necessary. The expected payoffs of the resident $\pi_{\mathrm{res}}$ and the mutant $\pi_{\mathrm{mut}}$ for $T'$ rounds are

$$\begin{cases} \pi_{\mathrm{res}} &= -cT/2 + (b - c)(T' - T)/2 \\ \pi_{\mathrm{mut}} &= 0. \end{cases} \tag{24}$$

                                        

Therefore, the requirement $\pi_{\text{res}} > \pi_{\text{mut}}$ reduces to the following condition on $b/c$:

$$R_1(B, G, C) > R_1(B, G, D)$$

$$\text{and}$$

$$\frac{b}{c} > \frac{R_1(B, G, C) + R_2(G, B, D)}{R_1(B, G, C) - R_1(B, G, D)}. \tag{25}$$

(*ii*) Next, we explore the context that both the donor and the recipient are good, ($G$, $G$). If the norm is to be self-cooperative, Eq (21) requires a resident donor to cooperate, $P(G, G) = 1$. A mutant donor instead chooses to defect, $P'(G, G) = 0$. For the resident norm to be stable, $R_1(G, G, D) < R_1(G, G, C)$ is necessary, for otherwise mutants save the cooperation cost without a penalty to their reputation. Now to compute the payoffs, we note that a good resident donor with $P(G, G) = 1$ pays an immediate cost $c$. In return, by Eq (20), this donor maintains its good reputation. In the subsequent interactions, this donor obtains on average $(b - c)/2$ for each round. The expected payoff over the next $T$ rounds is thus $\pi_{\text{res}} = (b - c)T/2 - c$. On the other hand, consider a mutant donor who defects, and follows the resident's action rule $P$ thereafter. Because this mutant does not receive a benefit for the next $T$ rounds when its reputation gets bad by Eq (22), its payoff over the next $T$ rounds is $\pi_{\text{mut}} = [1 - R_1(G, G, D)](-cT)/2 + R_1(G, G, D)(b - c)T/2$. By comparing these expected payoffs, we conclude that $P(G, G) = 1$ is optimal if and only if

$$R_1(G, G, C) > R_1(G, G, D)$$

$$\text{and}$$

$$\frac{b}{c} > \frac{R_1(B, G, C) + R_2(G, B, D)}{R_1(G, G, C) - R_1(G, G, D)}. \tag{26}$$

(*iii*) Now consider the context that a good donor interacts with a bad recipient, ($G$, $B$). For this context, Eq (22) requires the resident to apply an action rule with $P(G, B) = 0$. A deviating mutant uses $P'(G, B) = 1$. A resident donor's expected payoff over the following $T$ rounds is $R_1(G, B, D)(b - c)T/2 + [1 - R_1(G, B, D)](-c)T/2$. In contrast, a mutant gets $R_1(G, B, C)(b - c)T/2 + [1 - R_1(G, B, C)](-c)T/2 - c$. The resident has the higher payoff if and only if

$$R_1(G, B, C) \leq R_1(G, B, D)$$

$$\text{or}$$

$$\frac{b}{c} < \frac{R_1(B, G, C) + R_2(G, B, D)}{R_1(G, B, C) - R_1(G, B, D)}. \tag{27}$$

(*iv*) Finally, we determine the optimal action when both players are bad, ($B$, $B$). Because $h^* = 1$, the chance that the donor is matched with another $B$ player later on is negligible. Thus, we consider the long-term payoffs over $T$ rounds in which the donor is matched with $G$ players in all subsequent rounds. By cooperating in the first round, the donor gets $-c + R_1(B, B, C)(b - c)T/2 + [1 - R_1(B, B, C)](-c)T/2$. By defecting, the donor gets $R_1(B, B, D)(b - c)T/2 + [1 - R_1(B, B, D)](-c)T/2$. Therefore, $P(B, B) = 1$ is optimal if and only if

$$R_1(B, B, C) > R_1(B, B, D)$$

$$\text{and}$$

$$\frac{b}{c} > \frac{R_1(B, G, C) + R_2(G, B, D)}{R_1(B, B, C) - R_1(B, B, D)}. \tag{28}$$

Otherwise $P(B, B) = 0$ is optimal.

                    

We can summarize the above observations as follows. A norm with $P(B, G) = 1$ is a CESS if and only if the following conditions are met:

$$
\begin{cases}
R_1(G, G, C) = 1 \\
R_2(G, G, C) = 1 \\
R_1(G, B, D) + R_2(G, B, D) + R_1(B, G, C) + R_2(B, G, C) > 2 \\
P(G, G) = 1 \\
P(G, B) = 0 \\
P(B, G) = 1 \\
P(B, B) = \begin{cases} 1 & \text{if Eq. (28) holds} \\ 0 & \text{otherwise} \end{cases} \\
R_1(G, G, D) < 1 \\
R_1(B, G, C) > R_1(B, G, D) \\
\dfrac{b}{c} > \max\left\{ \dfrac{R_1(B, G, C) + R_2(G, B, D)}{R_1(G, G, C) - R_1(G, G, D)}, \dfrac{R_1(B, G, C) + R_2(G, B, D)}{R_1(B, G, C) - R_1(B, G, D)} \right\} \\
R_1(G, B, C) \leq R_1(G, B, D) \quad \text{or} \quad \dfrac{b}{c} < \dfrac{R_1(B, G, C) + R_2(G, B, D)}{R_1(G, B, C) - R_1(G, B, D)}
\end{cases}
\tag{29}
$$

Based on the above conditions, we conclude that social norms are particularly likely to satisfy the conditions of a CESS if they have the following properties.

1. By Eq (26), the norm should satisfy $R_1(G, G, C) \gg R_1(G, G, D)$. That is, among good players, cooperation needs to substantially increase the donors' chance to maintain a good reputation.

2. By Eq (25), the norm should satisfy $R_1(B, G, C) \gg R_1(B, G, D)$. When a bad donor interacts with a good recipient, cooperation needs to increase the chance that the donor's reputation recovers.

3. By Eq (27), the norm should satisfy $R_1(G, B, D) \gg R_1(G, B, C)$. Good players should be incentivized to withhold cooperation from ill-reputed players.

4. By Eqs (25) and (26), $R_2(G, B, D)$ should be sufficiently small. That is, it should be difficult for bad players to passively recover their reputation as a recipient.

**Norms with $P(B, G) = 0$.** Next, we assume that bad players are supposed to defect against good recipients. We proceed in the same way as before, by checking all possible contexts in which a donor is to make a decision.

(*i*) To start with, consider the context in which a bad donor is matched with a good recipient, $(B, G)$. By assumption, a resident cooperates with probability $P(B, G) = 0$, whereas the mutant has $P'(B, G) = 1$. We already compared the respective payoffs in the previous section; $P(B, G) = 0$ is optimal if and only if

$$
R_1(B, G, C) \leq R_1(B, G, D)
$$
$$
\text{or}
$$
$$
\frac{b}{c} < \frac{R_1(B, G, C) + R_2(G, B, D)}{R_1(B, G, C) - R_1(B, G, D)}.
\tag{30}
$$

(*ii*) Next, we assume that both players have a good reputation, $(G, G)$. By Eq (21), a resident donor is required to cooperate, $P(G, G) = 1$, which implies that a mutant would defect, $P'(G, G) = 0$. (Again we assume that compared to the resident, the mutant deviates once and then behaves identical to the resident later on). A resident pays an immediate cost $c$ but keeps a good reputation for the following $T'$ rounds, yielding an expected payoff of $-c + R_1(G, G, C)(b - c)T'/2$. In contrast, the mutant may get a bad reputation after the first defection. In that case, the mutant pays no cost, $P(B, G) = 0$, and keeps a bad reputation for the following $T'$ rounds. The resulting payoff is $[1 - R_1(G, G, D)]T' \times 0 + R_1(G, G, D)(b - c)T'/2$. This payoff is smaller than the resident's payoff if and only if

$$R_1(G, G, C) > R_1(G, G, D)$$
$$\text{and}$$
$$\frac{b}{c} > 1 + \frac{R_1(B, G, D) + R_2(G, B, D)}{R_1(G, G, C) - R_1(G, G, D)}. \tag{31}$$

(*iii*) Now we consider interactions among a good donor and a bad recipient, $(G, B)$. By Eq (22), residents need to defect, $P(G, B) = 0$. This in turn implies that mutants would cooperate, $P'(G, B) = 1$. For a defecting resident, the expected payoff for the following $T'$ round is $R_1(G, B, D)(b - c)T'/2 + [1 - R_1(G, B, D)]T' \times 0$. In contrast, a cooperating mutant gets $-c + R_1(G, B, C)(b - c)T'/2 + [1 - R_1(G, B, C)]T' \times 0$. The resident payoff is greater if and only if

$$R_1(G, B, C) \leq R_1(G, B, D)$$
$$\text{or}$$
$$\frac{b}{c} < 1 + \frac{R_1(B, G, D) + R_2(G, B, D)}{R_1(G, B, C) - R_1(G, B, D)}. \tag{32}$$

(*iv*) To determine the optimal $P(B, B)$ we consider a bad donor who interacts with another bad recipient. Moreover, because $h^* = 1$, we suppose the donor subsequently only meets good recipients for $T'$ rounds. A player with $P(B, B) = 1$ gets $-c + R_1(B, B, C)(b - c)T'/2$, In contrast, a player with $P(B, B) = 0$ gets $R_1(B, B, D)(b - c)T'/2$. Therefore, $P(B, B) = 1$ is optimal if and only if

$$R_1(B, B, C) > R_1(B, B, D)$$
$$\text{and}$$
$$\frac{b}{c} > 1 + \frac{R_1(B, G, D) + R_2(G, B, D)}{R_1(B, B, C) - R_1(B, B, D)}. \tag{33}$$

Otherwise $P(B, B) = 0$ is optimal.

To summarize, a norm with $P(B, G) = 0$ is a CESS if and only if

$$
\begin{cases}
R_1(G, G, C) = 1 \\
R_2(G, G, C) = 1 \\
R_1(G, B, D) + R_2(G, B, D) + R_1(B, G, D) + R_2(B, G, D) > 2 \\
P(G, G) = 1 \\
P(G, B) = 0 \\
P(B, G) = 0 \\
P(B, B) = \begin{cases} 1 & \text{if Eq. (33) holds} \\ 0 & \text{otherwise} \end{cases} \\
R_1(G, G, D) < 1 \\
\dfrac{b}{c} > 1 + \dfrac{R_1(B, G, D) + R_2(G, B, D)}{R_1(G, G, C) - R_1(G, G, D)} \\
R_1(B, G, C) \le R_1(B, G, D) \quad \text{or} \quad \dfrac{b}{c} < \dfrac{R_1(B, G, C) + R_2(G, B, D)}{R_1(B, G, C) - R_1(B, G, D)} \\
R_1(G, B, C) \le R_1(G, B, D) \quad \text{or} \quad \dfrac{b}{c} < 1 + \dfrac{R_1(B, G, D) + R_2(G, B, D)}{R_1(G, B, C) - R_1(G, B, D)}
\end{cases}
\tag{34}
$$

The general properties that make such social norms particularly likely to be stable are similar to the previous case of norms with $P(B, G) = 1$. The only exception occurs in the second item, which needs to be replaced by the following:

2'. By Eq (30) the norm should satisfy $R_1(B, G, C) \ll R_1(B, G, D)$. That is, bad players should recover a good reputation even if they defect against a good player.

This second rule differs from the previous case when $P(B, G) = 1$. However, in either case, the rules are consistent with the action prescribed by $P(B, G)$.

**Error sensitivity.**   Lastly, we discuss how sensitive CESS norms are with respect to implementation and assessment errors. For non-zero error rates $\mu_e$, $\mu_{a1}$, $\mu_{a2}$, the cooperation level $p_{res \to res}$ decreases from 1. We interpret the magnitude of this decrease as the norm's error sensitivity. Based on a heuristic argument, we derive the following expression for the Taylor expansion of $(1 - p_{res \to res})$ with respect to the error rates as

$$
1 - p_{res \to res} \approx (1 + [2 - R_1(G, G, D) - R_2(G, G, D)]\chi)\,\mu_e + \chi\,\mu_{a1} + \chi\,\mu_{a2},
\tag{35}
$$

where

$$
\chi \equiv \begin{cases}
\dfrac{1}{R_1(G, B, D) + R_2(G, B, D) + R_1(B, G, C) + R_2(B, G, C) - 2} & \text{if } P(B, G) = 1 \\[2mm]
\dfrac{2}{R_1(G, B, D) + R_2(G, B, D) + R_1(B, G, D) + R_2(B, G, D) - 2} & \text{if } P(B, G) = 0
\end{cases}
\tag{36}
$$

Below, we provide an intuition for these Eqs (35) and (36). To this end, we assume errors are sufficiently rare, such that after an initial error no further errors occur until individual reputations are recovered. Let us consider the case of assessment errors first. When an assessment error occurs, a bad player is introduced into a population of otherwise good players. This player causes $\chi$ times of defections in total, as shown in the following. It takes $T$ or $T'$ rounds until the bad player recovers a good reputation when $P(B, G) = 1$ and $P(B, G) = 0$, respectively.

For $P(B, G) = 1$, defection occurs when the bad player is chosen as the recipient. Therefore, on average there are $T/2$ rounds in which the bad player causes a defection. However, during $T$ rounds, interactions with the bad player may also lead other players to have a bad reputation. When the bad player is the recipient of the game, the donor becomes bad with probability $1 - R_1(G, B, D)$. When the bad player is the donor of the game, the recipient becomes bad with probability $1 - R_2(B, G, C)$. Each of these events happens $T/2$ times. Each new bad player may cause further good players to lose their good reputation. Overall, the total number of bad players initiated by a single bad player can be computed as the result of a geometric series,

$$\frac{1}{1 - [2 - R_1(G, B, D) - R_2(B, G, C)]T/2}. \tag{37}$$

As a consequence, the total number of defections caused by a single assessment error is

$$\frac{1}{1 - [2 - R_1(G, B, D) - R_2(B, G, C)]T/2} \times \frac{T}{2}. \tag{38}$$

On the other hand, for $P(B, G) = 0$ and initially one bad player, there are on average $T'$ rounds with defection until the bad player's reputation is recovered (because defection occurs independent of whether the bad player is chosen as the donor or recipient). The total number of bad players caused by a single bad player is

$$\frac{1}{1 - [2 - R_1(G, B, D) - R_2(B, G, D)]T'/2}. \tag{39}$$

As a result, the total number of defections caused by a single assessment error is

$$\frac{1}{1 - [2 - R_1(G, B, D) - R_2(B, G, D)]T'/2} \times T'. \tag{40}$$

By using the explicit expressions for $T$ and $T'$ in Eq (23), we obtain Eq (36). Interestingly, in each case the denominator of $\chi$ is equal to $-\frac{dh}{dh}\big|_{h=1}$, see Eq (20). This is not a coincidence. After all, both quantities characterize how quickly $h$ recovers to $h = 1$, and in each case $\overline{R}_1(G, B) + \overline{R}_2(G, B) + \overline{R}_1(B, G) + \overline{R}_2(B, G) > 2$ is required for recovery to occur.

Implementation errors are more complicated to describe. An implementation error causes a single defection additionally to bad reputations because of the erroneous action. On average, a single defection results in $[2 - R_1(G, G, D) - R_2(G, G, D)]$ bad players. Each of these bad players causes $\chi$ additional defections. Therefore, the total number of defections caused by a single implementation error is $1 + [2 - R_1(G, G, D) - R_2(G, G, D)]\chi$.

By summing up the effects of these errors, Eq (35) gives the overall reduction in the cooperation level. We verified that Eq (35) agrees well with the numerical results for various social norms unless the denominators of $\chi$ are too close to 0, where the first-order approximation is no longer valid.

The above discussion shows the fundamental tradeoff between sensitivity against errors and evolutionary stability. For $P(B, G) = 1$, the common terms $R_1(B, G, C)$ and $R_2(G, B, D)$ appear in the denominators of $\chi$ in Eq (36) and the numerators of the lower bound of $b/c$ in Eq (29). Namely, by controlling these terms, we can decrease either the lower $b/c$ bound or the error sensitivity, but not both. Same is true for $P(B, G) = 0$ as there are common terms $R_1(B, G, D)$ and $R_2(G, B, D)$ in Eqs (36) and (34). It indicates that the CESS norms are more sensitive to errors when the norms are evolutionary stable for broader ranges of $b/c$. We will see this tradeoff in the subsequent section.

## A discussion of several special cases

**Deterministic norms without updating the recipient.**  To explore the above results in more detail, we first consider the most simple class of social norms. We assume that the social norm is deterministic, $P(X, Y) \in \{0, 1\}$, $R_i(X, Y, A) \in \{0, 1\}$. In addition, we assume the recipient's reputation is kept constant, $R_2(X, Y, A) = \delta_{Y,G}$.

Let us first consider the case when $P(B, G) = 1$, as described in Eq (29). Because of the condition $R_1(G, B, D) + R_2(G, B, D) + R_1(B, G, C) + R_2(B, G, C) > 2$, we conclude that $R_1(G, B, D) = 1$ and $R_1(B, G, C) = 1$. Next, because of $R_1(B, G, C) > R_1(B, G, D)$, it follows that $R_1(B, G, D) = 0$. By requiring the lower bound of $b/c$ to be a finite value, we get $R_1(G, G, D) = 0$. The upper bound of $b/c$ goes to infinity since $R_1(G, B, D) = 1$. To summarize, we obtain the following conditions that are consistent with the definition of the leading eight:

$$\begin{cases} R_1(G, G, C) = 1 \\ R_1(G, G, D) = 0 \\ R_1(G, B, D) = 1 \\ R_1(B, G, C) = 1 \\ R_1(B, G, D) = 0 \\ P(G, G) = 1 \\ P(G, B) = 0 \\ P(B, G) = 1 \end{cases} \tag{41}$$

The three remaining values of $R_1(G, B, C)$, $R_1(B, B, C)$, $R_1(B, B, D)$ are arbitrary, and $P(B, B)$ is determined according to Eq (28). When these conditions are met, the norms are CESS norms for $b/c > 1$. Thus, we get eight CESS norms in total, which coincide with the leading eight, as shown in Table 1.

Next, we consider the case when $P(B, G) = 0$. From Eq (34), the following are required to construct the deterministic CESS norms:

$$\begin{cases} R_1(G, G, C) = 1 \\ R_1(G, G, D) = 0 \\ R_1(G, B, D) = 1 \\ R_1(B, G, D) = 1 \\ P(G, G) = 1 \\ P(G, B) = 0 \\ P(B, G) = 0 \end{cases} \tag{42}$$

There are four unspecified values $R_1(G, B, C)$, $R_1(B, G, C)$, $R_1(B, B, C)$, $R_1(B, B, D)$. Moreover, the value of $P(B, B)$ is determined according to Eq (33). Therefore, there are 16 norms in total, whose definitions are shown in Table 2. These norms become CESS for $b/c > 2$. Hereafter, we refer to them as "the secondary sixteen."

Among the secondary sixteen, S16 is particularly simple to describe. Here, the only instance in which a player obtains a bad reputation is when a good donor defects against a good recipient, $R_1(G, G, D) = 0$. In all other cases, the donor is assessed as good. In particular, a bad player's reputation is always reset to good the next time the player acts as a donor, irrespective of the player's action. According to this norm, good players who defect against other good players

**Table 2. The prescriptions of the secondary sixteen.** The format of the table is the same as that of Table 1. The columns that vary among the different norms are highlighted in bold. In the bottom row, the common prescriptions are shown. $P(B, B)$, indicated by †, is 1 if and only if $R_1(B, B, C) = 1$ and $R_1(B, B, D) = 0$ according to Eq (33). All norms in this table are CESS for $b/c > 2$.

| | (G, G) | | | (G, B) | | | (B, G) | | | (B, B) | | |
|---|---|---|---|---|---|---|---|---|---|---|---|---|
| | $P$ | $R_1(C)$ | $R_1(D)$ | $P$ | $R_1(C)$ | $R_1(D)$ | $P$ | $R_1(C)$ | $R_1(D)$ | $P$ | $R_1(C)$ | $R_1(D)$ |
| S1 | 1 | 1 | 0 | 0 | **0** | 1 | 0 | **0** | 1 | **0** | **0** | **0** |
| S2 | 1 | 1 | 0 | 0 | **0** | 1 | 0 | **0** | 1 | **0** | **0** | **1** |
| S3 | 1 | 1 | 0 | 0 | **0** | 1 | 0 | **0** | 1 | **1** | **1** | **0** |
| S4 | 1 | 1 | 0 | 0 | **0** | 1 | 0 | **0** | 1 | **0** | **1** | **1** |
| S5 | 1 | 1 | 0 | 0 | **0** | 1 | 0 | **1** | 1 | **0** | **0** | **0** |
| S6 | 1 | 1 | 0 | 0 | **0** | 1 | 0 | **1** | 1 | **0** | **0** | **1** |
| S7 | 1 | 1 | 0 | 0 | **0** | 1 | 0 | **1** | 1 | **1** | **1** | **0** |
| S8 | 1 | 1 | 0 | 0 | **0** | 1 | 0 | **1** | 1 | **0** | **1** | **1** |
| S9 | 1 | 1 | 0 | 0 | **1** | 1 | 0 | **0** | 1 | **0** | **0** | **0** |
| S10 | 1 | 1 | 0 | 0 | **1** | 1 | 0 | **0** | 1 | **0** | **0** | **1** |
| S11 | 1 | 1 | 0 | 0 | **1** | 1 | 0 | **0** | 1 | **1** | **1** | **0** |
| S12 | 1 | 1 | 0 | 0 | **1** | 1 | 0 | **0** | 1 | **0** | **1** | **1** |
| S13 | 1 | 1 | 0 | 0 | **1** | 1 | 0 | **1** | 1 | **0** | **0** | **0** |
| S14 | 1 | 1 | 0 | 0 | **1** | 1 | 0 | **1** | 1 | **0** | **0** | **1** |
| S15 | 1 | 1 | 0 | 0 | **1** | 1 | 0 | **1** | 1 | **1** | **1** | **0** |
| S16 (Forgiver) | 1 | 1 | 0 | 0 | **1** | 1 | 0 | **1** | 1 | **0** | **1** | **1** |
| common | 1 | 1 | 0 | 0 | * | 1 | 0 | * | 1 | † | * | * |

are identified, yet they are forgiven unconditionally in the next round. Thus one may refer to the norm S16 as "Forgiver".

The leading eight and the secondary sixteen are the only CESS norms when the rules are deterministic and the recipient's reputation is kept constant. It is impossible to construct another CESS norm even for a higher $b/c$. We numerically conducted a comprehensive enumeration and verified these theoretical predictions (see Appendix for the details of the numerical methods).

**Deterministic norms updating the recipient's reputation.** In a next step, we explore the space of all deterministic norms, including those for which the recipient's reputation is updated. In total, there are $2^{20} = 1,048,576$ deterministic norms. Considering the symmetry concerning the swap of $G$ and $B$, the number of independent norms is 524, 800. Out of those, we find there are 2,944 CESS, see Table 3 for a comprehensive list. This list contains several distinct classes of social norms, which we describe in the following.

The first three rows in Table 3 are variants of the leading eight. For these norms, the rules $P$ and $R_1$ are identical to the leading eight. Additionally, $R_2(G, G, C) = 1$ and $R_2(G, B, D) + R_2(B, G, C) > 1$ are necessary and sufficient to obtain CESS. Each of these rows has five arbitrary entries, highlighted with an asterisk $*$. Therefore we can construct $2^5 = 32$ different $R_2$ rules, which in total, results in $8 \times 32 \times 3 = 768$ norms being in this class. Please note that the lower bound of $b/c$ depends on $R_2(G, B, D)$. When $R_2(G, B, D) = 1$, the lower bound increases by one compared to the cases with $R_2(G, B, D) = 0$.

The next three rows in Table 3 are variants of the secondary sixteen. Here, $P$ and $R_1$ are the same as those of the secondary sixteen. In addition, $R_2(G, G, C) = 1$ and $R_2(G, B, D) + R_2(B, G, D) > 1$ need to hold. This leaves five entries in each row unspecified. In total, there are $16 \times 2^5 \times 3 = 1,536$ CESS norms. Again, the lower bound of $b/c$ increases by one if $R_2(G, B, D) = 1$, compared to cases with $R_2(G, B, D) = 0$.

**Table 3. The CESS norms when the recipient's reputation is updated.** The leftmost column indicates the pair of $P$ and $R_1$. $L1, \ldots, L8$ and $S1, \ldots, S16$ mean that the prescriptions are the same as those of the leading eight and the secondary sixteen, respectively. $L'$, $S'$, and $S''$ are the variants of the leading eight and the secondary sixteen, whose definitions are given in Table 4. The middle columns show the rules of $R_2$. The asterisks indicate that the entry is arbitrary. The third column from the right shows the range of $b/c$ for which the norms are CESS. The rightmost two columns show the number of CESS norms, and the expected number $\chi$ of defections caused by a single bad player, respectively. Columns in which there is variation are highlighted in bold.

| $\{P, R_1\}$ | $(G, G)$ | | $(G, B)$ | | $(B, G)$ | | $(B, B)$ | | | # of norms | $\chi$ |
|---|---|---|---|---|---|---|---|---|---|---|---|
| | $R_2(C)$ | $R_2(D)$ | $R_2(C)$ | $R_2(D)$ | $R_2(C)$ | $R_2(D)$ | $R_2(C)$ | $R_2(D)$ | | | |
| $L1, \ldots, L8$ | 1 | * | * | **0** | **1** | * | * | * | $b/c > 1$ | 256 | 1 |
| $L1, \ldots, L8$ | 1 | * | * | **1** | **0** | * | * | * | $b/c > 2$ | 256 | 1 |
| $L1, \ldots, L8$ | 1 | * | * | **1** | **1** | * | * | * | $b/c > 2$ | 256 | 1/2 |
| $S1, \ldots, S16$ | 1 | * | * | **0** | * | **1** | * | * | $b/c > 2$ | 512 | 2 |
| $S1, \ldots, S16$ | 1 | * | * | **1** | * | **0** | * | * | $b/c > 3$ | 512 | 2 |
| $S1, \ldots, S16$ | 1 | * | * | **1** | * | **1** | * | * | $b/c > 3$ | 512 | 1 |
| $L'$ | 1 | * | * | **1** | **1** | * | * | * | $b/c > 2$ | 128 | 1 |
| $S'$ | 1 | * | * | **1** | * | **1** | * | * | $b/c > 2$ | 256 | 2 |
| $S''$ | 1 | * | * | **1** | * | **1** | * | * | $b/c > 3$ | 256 | 2 |

In addition to the above-mentioned norms, there are norms that differ in their $P$ and $R_1$ from the leading eight and the secondary sixteen. These norms comprise the last three rows of Table 3. In the row indicated by $L'$, the values of $R_1$ are similar to the leading eight, see Table 4. The major difference to the leading eight corresponds to the entry $R_1(G, B, D) = 0$. According to $L'$, defecting against a bad player is not justified. Another minor difference is that $R_1(G, B, C) = 0$, while this value is arbitrary in the leading eight. There are two and five arbitrary entries in $R_1$ and in $R_2$, respectively. Therefore, there are $2^2 \times 2^5 = 128$ CESS norms in this class of $L'$ norms. Again, the lower bound of $b/c$ increases by one because of $R_2(G, B, D) = 1$.

For the rows in Table 3 indicated by $S'$ and $S''$, the entries of $R_1$ are similar to the entries of the secondary sixteen. In each case, there is one entry that is different. In addition, there is one other entry, which is arbitrary for the secondary sixteen but fixed to 0 for $S'$ and $S''$. Because three and five entries in $R_1$ and $R_2$ are left unspecified, respectively, there are $2^3 \times 2^5 = 256$ CESS norms for each row. The lower bound of $b/c$ increases by one because $R_2(G, B, D) = 1$. However, for $S'$, the lower bound of $b/c$ decreases by one since $R_1(B, G, D) = 0$. Hence the norms in $S'$ are stable for $b/c > 2$, while the norms in $S''$ require $b/c > 3$.

With respect to the minimal $b/c$ ratio required for cooperation, we note that already the original leading-eight norms are optimal. However, other norms fare better with respect to a different metric, a norm's ability to correct errors. We illustrate this relationship in Fig 2. Here, the $y$-axis shows the minimum threshold for $b/c$ such that the social norm is a CESS. The $x$-axis shows a norm's error sensitivity. This sensitivity is defined as the number of defections that are triggered by an error, $(1 - p_{\text{res}\to\text{res}})$ normalized by the error rate $\mu (\equiv \mu_e = \mu_{a1} = \mu_{a2})$.

**Table 4. The definitions of $L'$, $S'$, and $S''$.** $L'$ is similar to one of the leading eight. The opposite prescription from the leading eight is $R_1(G, B, D) = 0$, which is highlighted in bold text. You can identify another difference in $R_1(G, B, C)$, denoted in italics. It is fixed to be 0 for $L'$ whereas it is arbitrary for the leading eight. $S'$ and $S''$ are variants of the secondary sixteen. Again, the entries opposite to the secondary sixteen are indicated in bold text while the fixed entries are indicated in italics. The asterisks $*$ indicate arbitrary entries and the daggers † are determined according to Eqs (28) or (33).

| type | $(G, G)$ | | | $(G, B)$ | | | $(B, G)$ | | | $(B, B)$ | | |
|---|---|---|---|---|---|---|---|---|---|---|---|---|
| | $P$ | $R_1(C)$ | $R_1(D)$ | $P$ | $R_1(C)$ | $R_1(D)$ | $P$ | $R_1(C)$ | $R_1(D)$ | $P$ | $R_1(C)$ | $R_1(D)$ |
| $L'$ | 1 | 1 | 0 | 0 | *0* | **0** | 1 | 1 | 0 | † | * | * |
| $S'$ | 1 | 1 | 0 | 0 | * | 1 | 0 | *0* | **0** | † | * | * |
| $S''$ | 1 | 1 | 0 | 0 | *0* | **0** | 0 | * | 1 | † | * | * |

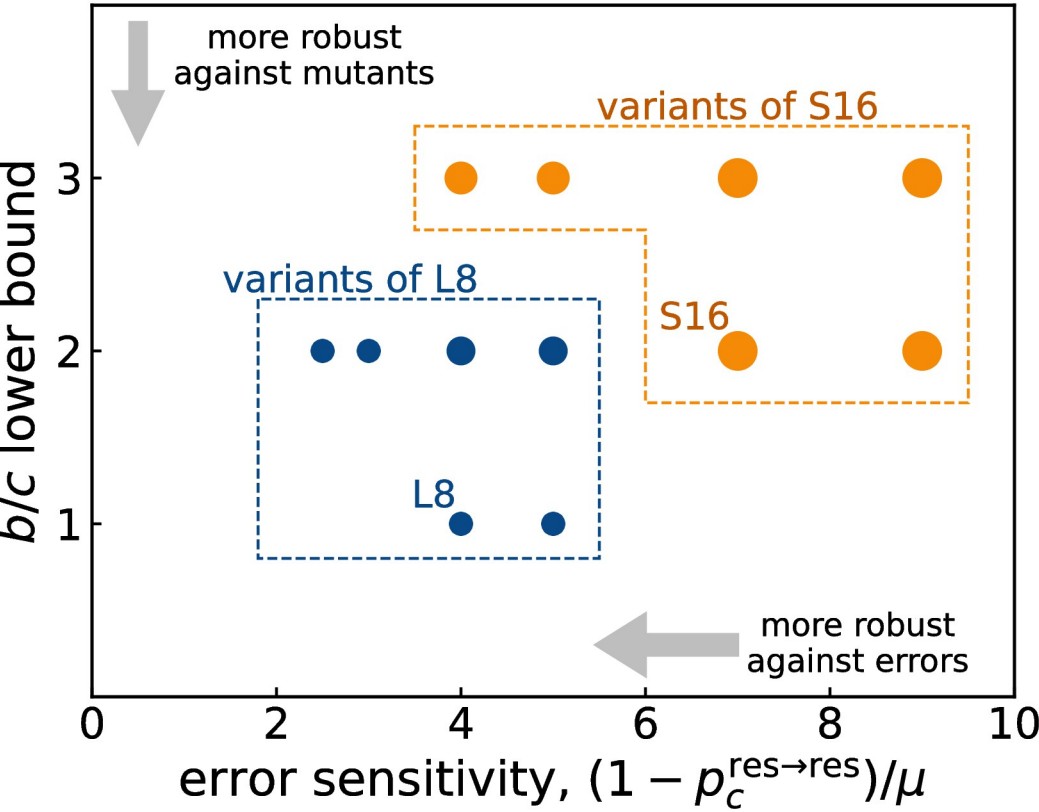

**Fig 2. Error sensitivity vs. the lower bound of the benefit-to-cost ratio.** Error sensitivity $(1 - p_{\text{res}\to\text{res}})/\mu$ and the lower bound of $b/c$ for the deterministic CESS norms in Table 3. Each point represents norms, and its size indicates the number of norms. The texts, "L8" and "S16", indicate where the leading eight and the secondary sixteen belong, respectively. The blue and orange points are the variants of the leading eight and the secondary sixteen, respectively.

The lower this error sensitivity, the better a social norm is able to correct errors. We confirmed that the number of defections scales linearly with the error rate for small enough error rates, $\mu \lesssim 10^{-2}$. Moreover, we confirmed that the error sensitivity is consistent with the Taylor expansion of $p_{\text{res}\to\text{res}}$ with respect to $\mu$.

As shown in Fig 2, the CESS fall into several discrete classes. Blue points represent variants of the leading eight, whereas orange points are variants of the secondary sixteen. The pure leading eight and secondary sixteen (those without recipient updating) have coordinates (4, 1) and (7, 2), respectively. Remarkably, there are norms at (5/2, 2) and (3, 2) with an even lower error sensitivity. These are the norms in the third row of Table 3. In terms of $P$ and $R_1$ they coincide with the leading eight. In addition, they satisfy $R_2(G, G, C) = 1$, $R_2(G, B, D) = 1$, and $R_2(B, G, C) = 1$. Compared to the leading eight, the key difference is

$$R_2(G, B, D) = 1. \tag{43}$$

This rule stipulates that bad recipients should be forgiven once they have been defected against. In this way, the rule ensures that bad reputations come with some disadvantage, but that this disadvantage is not lasting. The downside of this leniency is that such norms require larger values of $b/c$ to be stable. Overall, these results illustrate a fundamental tradeoff between a norm's stability against errors and its evolutionary stability against mutants.

**Table 5. The second-order CESS norms.** The second-order social norms among the deterministic CESS norms. The second-order norms are the norms whose rules do not depend on the donor's reputation. The rightmost two columns show the range of $b/c$ for which the norms are CESS, and $\chi$.

| $\{P, R_1\}$ | $(-, G)$ | | | | | $(-, B)$ | | | | | | $\chi$ |
|---|---|---|---|---|---|---|---|---|---|---|---|---|
| | $P$ | $R_1(C)$ | $R_1(D)$ | $R_2(C)$ | $R_2(D)$ | $P$ | $R_1(C)$ | $R_1(D)$ | $R_2(C)$ | $R_2(D)$ | | |
| $L3$ or $L6$ | 1 | 1 | 0 | 1 | * | 0 | * | 1 | * | 0 | $b/c > 1$ | 1 |
| $L3$ or $L6$ | 1 | 1 | 0 | 1 | * | 0 | * | 1 | * | 1 | $b/c > 2$ | 1/2 |
| $L'$ | 1 | 1 | 0 | 1 | * | 0 | 0 | 0 | * | 1 | $b/c > 2$ | 1 |

The difference between norms with coordinates (5/2, 2) and (3, 2) is caused by the entry $R_2(G, G, D)$. Norms at (5/2, 2) have $R_2(G, G, D) = 1$ whereas norms at (3, 2) have $R_2(G, G, D) = 0$. It is intuitive that a lower $R_2(G, G, D)$ leads to a higher error sensitivity especially when donors are subject to implementation errors.

Herein, we have studied all so-called third-order norms. According to these norms, assessments depend on a donor's action, and both the donor's and the recipient's previous reputation. However, we note that several of these norms can in fact be represented as simpler second-order norms. These norms still depend on the donor's action and on the recipient's previous reputation, but they are independent of the donor's previous reputation. Among the leading eight, $L3$ (Simple Standing) and $L6$ (Stern Judging) are second-order norms. When we take $R_2$ into account, there are more second-order CESS norms as shown in Table 5. These are variants of $L3$, $L6$, and $L'$. When $R_2(-, B, D) = 1$, the norms are CESS for $b/c > 2$, where '$-$' indicates that the donor's reputation is irrelevant. For each entry in the table, there are two wildcards. Therefore, there are $5 \times 2^2 = 20$ norms in total. Interestingly, none of the secondary sixteen is second-order; they all have $P(G, G) = 1$ and $P(B, G) = 0$.

**Stochastic norms.** Finally, we discuss some simple examples of stochastic norms. The first example is a stochastic variant of L2 to which we refer as $sL2$, as defined in Table 6. This norm differs in that $R_1(G, G, D) = p_1$, $R_1(G, B, D) = p_2$, and $R_1(B, G, C) = p_3$. These three probabilities can be interpreted as follows. The probability $1 - p_1$ is the probability that a defector is assigned a bad reputation. The smaller $p_1$, the less forgiving a norm is with respect to (possibly accidental) defections. The second probability $p_2$ is the extent to which a norm respects justified defections (when a good donor defects against a bad recipient). When $p_2 < 1$, the donor may get a bad reputation even though the donor follows the prescribed action rule $P(G, B)$. The third probability $p_3$ controls how easy it is for bad donors to recover a good reputation. If $p_3 < 1$ it may take several acts of cooperation until a bad donor is deemed as good. Norms with $p_2 < 1$ and $p_3 < 1$ share some similarities with previously described norms with ternary reputations [41]. According to our results, the stochastic norm $sL2$ is a CESS for $b/c > 1$ if and only if $1 - p_1 > p_3$ and $p_2 + p_3 > 1$. For example, the probabilities $(p_1, p_2, p_3) = (0.3, 0.5, 0.7)$ suffice these conditions.

There are tradeoffs between the error sensitivity and the evolutionary stability also for these stochastic norms. Fig 3 shows the error sensitivity and the lower bound of $b/c$ for $sL2$. Here, we

**Table 6. Two examples of stochastic norms.** Stochastic norms, $sL2$ and $sS1$, are stochastic variants of $L2$ and $S1$, respectively. The norm $sL2$ is CESS if and only if $p_1 < 1$ and $p_2 + p_3 > 1$ for $b/c > \max\{p_3/(1 - p_1), 1\}$. The norm $sS1$ is CESS if and only if $p_1 < 1$ and $p_2 + p_3 > 1$ for $b/c > 1 + p_3/(1 - p_1)$.

| | $(G, G)$ | | | $(G, B)$ | | | $(B, G)$ | | | $(B, B)$ | | |
|---|---|---|---|---|---|---|---|---|---|---|---|---|
| | $P$ | $R_1(C)$ | $R_1(D)$ | $P$ | $R_1(C)$ | $R_1(D)$ | $P$ | $R_1(C)$ | $R_1(D)$ | $P$ | $R_1(C)$ | $R_1(D)$ |
| $sL2$ | 1 | 1 | $p_1$ | 0 | 0 | $p_2$ | 1 | $p_3$ | 0 | 1 | 1 | 0 |
| $sS1$ | 1 | 1 | $p_1$ | 0 | 0 | $p_2$ | 0 | 0 | $p_3$ | 0 | 0 | 0 |

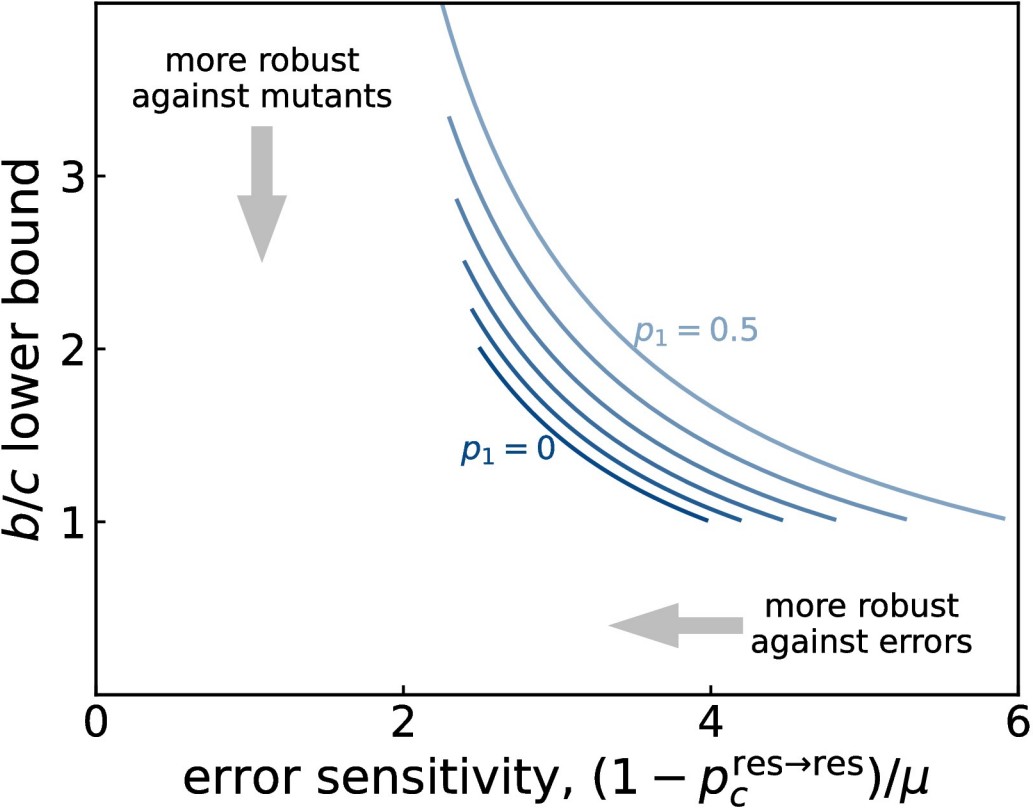

**Fig 3. Error sensitivity vs. the lower bound of the benefit-to-cost ratio for stochastic norms.** Error sensitivity $(1 - p_{\text{res}\rightarrow\text{res}})/\mu$ and the lower bound of $b/c$ for the stochastic CESS norms $sL2$. Each curve shows the tradeoff relationship for different $p1 \in [0, 0.5]$. While fixing $p_2 = 1$, we change $p_3 \in [1 - p_1, 1]$ and $p_4 \in [0, 1]$.

fix $p_2 = 1$ and change $p_1$ and $p_3$ in the range satisfying the conditions Eq (29). We also control $p_4 \equiv R_2(G, B, D)$. In this particular case, we have simple analytic forms from Eqs (29) and (35): the lower $b/c$ bound is $(p_3 + p_4)/(1 - p_1)$ and the error sensitivity is $1 + (3 - p_1)/(p_3 + p_4)$. As shown in the figure, we cannot simultaneously improve the error sensitivity and the lower bound of $b/c$.

As another example, we consider a stochastic norm derived from $S1$, one of the secondary sixteen. This norm $sS1$, defined in Table 6, differs from S1 in $R_1(G, G, D) = p_1$, $R_1(G, B, D) = p_2$, and $R_1(B, G, D) = p_3$. It is a CESS when $p_2 + p_3 > 1$, $p_1 < 1$ and $b/c > 1 + p_3/(1 - p_1)$. This example illustrates that stochasticity can be useful to extend the range of $b/c$ for which stable cooperation can be maintained. For instance, choosing the value $(p_1, p_2, p_3) = (0, 1, \epsilon)$ leads to $b/c > 1 + \epsilon$, which is a weaker condition than S1's threshold $b/c > 2$. The smaller $p_3$ becomes, the more time it takes a bad player to recover a good reputation. This reduces the incentive to defect, resulting in a CESS even for comparably small benefits of cooperation. With respect to the critical $b/c$ ratio, $sS1$ becomes (approximately) as powerful as the leading eight. On the other hand, there also exists a disadvantage of being vulnerable to errors.

In addition to these two examples, we also characterize all stochastic second-order CESS. Since these norms do not depend on the donor's reputation, they only include variants of the leading eight, as discussed before. Considering Eq (29), the second-order stochastic norms are

CESS if and only if

$$
\begin{cases}
P(-,G) = 1 \\
P(-,B) = 0 \\
R_1(-,G,C) = 1 \\
R_2(-,G,C) = 1 \\
R_1(-,B,D) + R_2(-,B,D) > 0 \\
R_1(-,G,D) < 1 \\
\dfrac{b}{c} > \dfrac{1 + R_2(-,B,D)}{1 - R_1(-,G,D)} \\
R_1(-,B,C) \le R_1(-,B,D) \quad \text{or} \quad \dfrac{b}{c} < \dfrac{1 + R_2(-,B,D)}{R_1(-,B,C) - R_1(-,B,D)}
\end{cases}
, \tag{44}
$$

If we further require that these norms work for $b/c > 1$, these conditions further simplify to

$$
\begin{cases}
P(-,G) = 1 \\
P(-,B) = 0 \\
R_1(-,G,C) = 1 \\
R_1(-,G,D) = 0 \\
R_1(-,B,D) > 0 \\
R_1(-,B,C) \le R_1(-,B,D) \\
R_2(-,G,C) = 1 \\
R_2(-,B,D) = 0
\end{cases}
. \tag{45}
$$

The remaining two entries $R_2(-,G,D)$ and $R_2(-,B,C)$ can take arbitrary values. We can summarize these rules as follows. First, when interacting with a good recipient, donors should cooperate. If they do so, donors should obtain a good reputation, and otherwise a bad reputation: $P(-,G) = 1$, $R_1(-,G,C) = 1$, $R_1(-,G,D) = 0$. Second, when interacting with a bad recipient, donors should defect. Defecting donors should obtain a good reputation with positive probability, whereas cooperating donors should be less likely to receive a good reputation: $P(-,B) = 0$, $R_1(-,B,D) > 0$, $R_1(-,B,C) \le R_1(-,B,D)$. Third, good recipients keep their reputation when they receive cooperation, whereas bad recipients keep their reputation when they receive defection, $R_2(-,G,C) = 1$, $R_2(-,B,D) = 0$.

After describing the second-order norms, we can even further constrain the norms such that donors are only assessed according to their actions, and that the recipients' reputations are not updated at all. These are the so-called first-order norms. According to Eq (44), it is straightforward to show that first-order norms cannot be CESS because the lower and the upper bound of $b/c$ coincide. However, if we allow (not strict) Nash equilibria (cases where

$\pi_{\mathrm{res}} \geq \pi_{\mathrm{mut}}$), we obtain the following conditions,

$$
\begin{cases}
P(-, G) = 1 \\
P(-, B) = 0 \\
R_1(-, -, C) = 1 \\
R_1(-, -, D) = 1 - \dfrac{c}{b}
\end{cases}
. \tag{46}
$$

Interestingly, this norm coincides with the "generous scoring" rule, which was previously proposed in a private reputation model [43]. In our general theoretical framework, we naturally recover this norm when we require the norm to be of first order.

## Discussion

In this paper, we extend the classical model of indirect reciprocity with public reputations in two directions. First, in addition to updating the reputations of (active) donors, we also allow the reputation of (passive) recipients to be re-evaluated. With this model of dual reputation updates, we can account for previous observations that victims of selfish actions tend to be seen in a more positive light than they perhaps deserve [42]. Norms that prescribe to re-evaluate the reputation of recipients have been previously studied by Marcus Frean and Stephen Marsland when they explore the relationship between indirect reciprocity and monetary transactions [45]. Here, we explore such norms in the more elementary framework of Ohtsuki and Iwasa [9, 10]. Second, we allow moral assessments to be stochastic. In particular, communities may forgive bad actions with a positive probability.

Our stochastic formulation naturally allows for explicit calculations of the expected payoffs. As a result, we can characterize all norms that are stable and that allow for full cooperation (CESS). Our analysis has two immediate implications for the previously considered special case of deterministic norms. First, if the recipients' reputations are kept constant, the leading eight and the secondary sixteen are the only CESS. Second, when we also allow updates in the recipients' reputations, the norms in Table 3 are all the deterministic CESS there are.

Our analysis highlights an interesting difference between the leading eight and the secondary sixteen. According to the leading eight, bad players need to cooperate when they are matched with good recipients to recover a good reputation. In contrast, according to the secondary sixteen, bad donors would defect, but they are forgiven anyway. In this sense, the secondary sixteen are more lenient. As a result, they require a larger benefit-to-cost ratio to be evolutionarily stable, $b/c > 2$. Because (costly) apologies are broadly observed in a wide variety of cultures, the leading eight certainly seem more natural. This in turn might suggest that human norms have evolved in environments with low benefit-to-cost ratios.

As we have seen in Fig 2, there can be intriguing effects when social norms allow people to re-evaluate the reputations of passive recipients. In particular, we find norms that are better able to correct errors than the traditionally studied leading eight. At the same time, however, the respective norms require a larger benefit of cooperation to be stable. This observation indicates that there is an interesting tradeoff between robustness with respect to errors on the one hand and evolutionary stability on the other hand (see also Fig 3). These findings could have important implications for private-information models [16–27]. In such models, members of a community form their opinions independently from each other. In particular, individuals may disagree on the reputations they assign to third parties. Previous work suggests that errors can naturally introduce such disagreements. These disagreements can further spread over time, which threatens the overall stability of indirect reciprocity. In such a context, norms with strong error-correcting properties seem particularly valuable.

## Methods

In this section, we describe the numerical method to confirm whether or not a norm is a CESS. Our method is based on previous studies [9, 13, 41]. For our numerical results, we use $\mu_e = \mu_{a1} = \mu_{a2} = 10^{-3}$ unless specified otherwise. For a given social norm with action rule $P(X, Y)$ and assessment rules $R_1(X, Y, A)$ and $R_2(X, Y, A)$, the algorithm proceeds as follows:

1. Calculate the average reputation $h^*$ using Eq (8). In case $A \approx 0$, this analytic expression may be numerically inaccurate. We use Newton's method to improve the accuracy.

2. Calculate the cooperation level $p_{\text{res}\rightarrow\text{res}}$ using Eq (9).

3. Reject the norm if $p_{\text{res}\rightarrow\text{res}} < p_{\text{th}}$, where $p_{\text{th}} = 0.98$ is a threshold for the self-cooperation rate.

4. Initialize the lower and the upper bounds of working $b/c$ as $L = 1$ and $U = \infty$.

5. Loop for 15 deterministic action rules $P' \neq P$.

    1. Calculate the mutant's reputation $H^*$ using Eq (13).

    2. Calculate the cooperation levels $p_{\text{mut}\rightarrow\text{res}}$ and $p_{\text{mut}\rightarrow\text{mut}}$ using Eqs (15) and (16).

    3. Using Eq (18), calculate the range of $b/c$ for which the resident norm is stable against $P'$. Update $L$ or $U$ accordingly.

6. If $L < U$, the norm is a CESS for $L < b/c < U$. Otherwise, the norm is not a CESS.

We excluded cases with $|U - L| < 10^{-3}$ or those with unrealistically large $L > 10$; these are edge cases originating from numerical errors.

To compare these numerical results with our analytical predictions in Table 3, we enumerated all possible deterministic norms and obtained all the CESS norms.

## Author Contributions

**Conceptualization:** Yohsuke Murase, Christian Hilbe.

**Formal analysis:** Yohsuke Murase.

**Investigation:** Yohsuke Murase, Christian Hilbe.

**Methodology:** Yohsuke Murase.

**Software:** Yohsuke Murase.

**Validation:** Christian Hilbe.

**Writing – original draft:** Yohsuke Murase.

**Writing – review & editing:** Yohsuke Murase, Christian Hilbe.

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
