## [Decision Letter · Decision Letter 0]

1 May 2023

Dear Dr. Murase,

Thank you very much for submitting your manuscript "Indirect reciprocity with stochastic rules" for consideration at PLOS Computational Biology. As with all papers reviewed by the journal, your manuscript was reviewed by members of the editorial board and by several independent reviewers. The reviewers appreciated the attention to an important topic. Based on the reviews, we are likely to accept this manuscript for publication, providing that you modify the manuscript according to the review recommendations.

Sincerely,

Feng Fu

Academic Editor

PLOS Computational Biology

Natalia Komarova

Section Editor

PLOS Computational Biology

Reviewer's Responses to Questions

**Comments to the Authors:**

Reviewer #1: This paper explores the topic of indirect reciprocity, introducing two new components for consideration: the updating of player reputations based on other players' actions, and the stochastic updating of reputations. The authors conduct a thorough investigation into the emergence of cooperative equilibria, with a particular focus on invasion barriers. The study reveals the existence of "Secondary Sixteen" norms, which are cooperative ESSes with higher invasion barriers than the classical "Leading Eight" norms; here the benefit of cooperation must be at least twice the cost of cooperation for these norms to be upheld. The authors present a comprehensive list of cooperative ESSes, including the minimum ratio between cooperation benefit and cost for each. The findings are significant, and I recommend the paper for acceptance with minor revisions:

1. In line 368 (and in Figure 2), it is assumed that mu_e = mu_a1=mu_a2. It would be interesting to see what influence each of the three different error rates has.

2. Similarly, in Subsection 5.3 the benefit-to-cost ratios are examined for some stochastic norms. Here, it would be interesting to see if the trade-off between error robustness and ESS robustness extends to the stochastic norms.

Reviewer #2: Thank you for the opportunity to review this manuscript.

This manuscript makes several substantial contributions to the literature on indirect reciprocity. First, it expands the standard third-order social norms, including the so-called "leading eight", to allow for the possibility that not only is the donor's reputation updated after an interaction but also the recipient's. Second, it identifies a class of deterministic, donor-updating social norms that is a cooperative evolutionary stable state (CESS) for b/c > 2, with b the benefit of cooperation and c the cost (in comparison to the leading eight, which are stable for b/c > 1); these differ from the "leading eight" insofar as bad recipients defect against good ones but are forgiven anyway. Third, it allows social norms to be stochastic, assigning reputations with certain probabilities (separate from errors). One surprising (and particularly ingenious) finding is that norms that are more robust against errors require higher b/c ratios to sustain stable cooperation. There are several interesting minor results, as well; for example, stochastic version of the S1 "secondary sixteen" norm is almost as "powerful" as the leading eight despite the counterintuitive "bad defects against good but is considered good nonetheless" rule.

This is all put together in a presentation that is methodical, thorough, and for the most part easy to follow. Sections 2 and 3 function as great outlines for the arguments used in the rest of the paper and provide a (badly needed) intuitive understanding of CESS norms, including the leading eight.

I offer some comments on issues that ought to be addressed.

One, the order of ideas in the abstract and introduction does not match the rest of the paper. Stochastic norms are named as the "first" extension of the standard indirect reciprocity norms; updating the recipient's reputation is named as the "second". But the actual presentation of these ideas is in the reverse order. In fact the paper's title feels a bit odd given that stochastic norms are only really zeroed in on briefly, in section 5.3. I would certainly recommend either re-ordering section 5 to match the order in which these ideas are named or simply swapping the order in which they are listed in the abstract and introduction. The authors could also consider a more general title that better captures the material they spend most of their time on (though I don't have any great ideas offhand; the current title has the advantage of being brief and catchy, and I imagine "Indirect reciprocity in which not only the donor's reputation is updated but also the recipient's" would be hard to compress into five or six words, so this should be taken as a weak suggestion).

Two, I would like to see a bit more motivation for stochastic norms. The exploration of stochastic norms is justified solely with reference to indirect reciprocity modeling, but surely there are real-world examples where norms can be modeled as stochastic–for example, averaging over small population differences in social norm, or assessment that depends on the context of an action in a way other than the reputation of the donor and recipient. Probably better motivations can be found. Indirect reciprocity has profound implications for the evolution of human moral systems, so some care to link this part of the paper to said moral systems would be appreciated. By contrast, updating the recipient's reputation is, in the text, well justified with reference to intuition and the literature.

Third, several of the leading eight are well known because they have catchy names (and because two of them are actually second-order). The authors have the opportunity to coin names for some of the new norms they've derived. Possibly it's hard to come up with punchy names for these; the rule "bad defects with good but is considered good anyway" is counterintuitive and maybe none of these "secondary sixteen" admit a good name. But the authors should at least try.

Overall, the content of the paper is intriguing and of great interest, but it would benefit from some reorganization and a bit more attention, not just to the modeling side of indirect reciprocity, but to its implications for human social systems.

Several extensions are possible (for example, a version of Fig. 2 that considers sensitivity to errors that occur at different rates), but my guess is that they would not provide much more intuition, so I don't think they're really necessary.

There is a typo in equation 4, on the second line (the h(t) (1 - h(t)) term); the first \\bar{R}_2 (B,G) wants to be \\bar{R}_1 (B,G). This error does not appear to have percolated anywhere else in the manuscript (for example equations 7 or 11), but I didn't check all the algebra, so the authors should go through it with a fine-toothed comb just in case.

**Have the authors made all data and (if applicable) computational code underlying the findings in their manuscript fully available?**

Reviewer #1: Yes

Reviewer #2: Yes

PLOS authors have the option to publish the peer review history of their article (what does this mean?). If published, this will include your full peer review and any attached files.

Reviewer #1: No

Reviewer #2: No

Figure Files:

Data Requirements:

Reproducibility:

References:

---

## [Decision Letter · Decision Letter 1]

13 Jun 2023

Dear Dr. Murase,

We are pleased to inform you that your manuscript 'Indirect reciprocity with stochastic and dual reputation updates' has been provisionally accepted for publication in PLOS Computational Biology.

Best regards,

Feng Fu

Academic Editor

PLOS Computational Biology

Natalia Komarova

Section Editor

PLOS Computational Biology

Reviewer's Responses to Questions

**Comments to the Authors:**

Reviewer #1: The authors have done a good job of implementing the comments. The added subsection and figure further emphasize their results, and I recommend the paper for publication.

Reviewer #2: My comments have been very well addressed. The new title better fits the content of the paper, the re-ordering of major results is now consistent throughout the paper, and the addition of the "forgiver" norm name is much appreciated. The justification for including stochasticity is improved. The authors could in principle consider adding a reference showing that people do indeed make moral judgments for stochastic reasons exogenous to the model, such as Leloup et al. (2018), but this is not essential.

Thank you very much for the opportunity to review this manuscript. I am happy to recommend publication.

**Have the authors made all data and (if applicable) computational code underlying the findings in their manuscript fully available?**

Reviewer #1: None

Reviewer #2: Yes

PLOS authors have the option to publish the peer review history of their article (what does this mean?). If published, this will include your full peer review and any attached files.

Reviewer #1: No

Reviewer #2: No

---

## [Editor Report · Acceptance letter]

27 Jun 2023

PCOMPBIOL-D-23-00337R1 

Indirect reciprocity with stochastic and dual reputation updates

Dear Dr Murase,

I am pleased to inform you that your manuscript has been formally accepted for publication in PLOS Computational Biology. Your manuscript is now with our production department and you will be notified of the publication date in due course.

With kind regards,

Zsuzsanna Gémesi
